# Inertial Measurement Units' Reliability for Measuring Knee Joint Angle during Road Cycling

Saša Obradović  and Sara Stančin *

Faculty of Electrical Engineering, University of Ljubljana, Tržaška c. 25, 1000 Ljubljana, Slovenia
* Correspondence: sara.stancin@fe.uni-lj.si; Tel.: +386-14768-939

**Abstract:** We explore the reliability of joint angles in road cycling obtained using inertial measurement units. The considered method relies on 3D accelerometer and gyroscope measurements obtained from two such units, appropriately attached to two adjacent body parts, measuring the angle of the connecting joint. We investigate the effects of applying a simple drift compensation technique and an error-state Kalman filter. We consider the knee joint angle in particular, and conduct two measurement trials, a 5 and a 20 minute one, for seven subjects, in a closed, supervised laboratory environment and use optical motion tracking system measurements as reference. As expected from an adaptive solution, the Kalman filter gives more stable results. The root mean square errors per pedalling cycle are below 3.2°, for both trials and for all subjects, implying that inertial measurement units are not only reliable for short measurements, as is usually assumed, but can be reliably used for longer measurements as well. Considering the accuracy of the results, the presented method can be reasonably extended to open, unsupervised environments and other joint angles. Implementing the presented method supports the development of cheaper and more efficient monitoring equipment, as opposed to using expensive motion tracking systems. Consequently, cyclists can have an affordable way of position tracking, leading to not only better bicycle fitting, but to the avoidance and prevention of certain injuries as well.

**Keywords:** inertial measurement units; sensor fusion; Kalman filter; cycling; joint biomechanics; knee joint angle

## 1. Introduction

Cycling biomechanics is a well-studied research field, usually addressing important questions of cycling posture analysis, injury prevention and performance improvement [1–6]. Guidelines for seat height, pedal position, pedalling rate, and force application have been presented [1–8].

Being directly linked to the activation of muscles and muscle groups, the knee joint angle has shown to have a great effect on one's technique and its possible further improvement [1]. The flexion of joints of the lower limbs in particular changes with the pedalling crank angle. It has been reported that cyclists during ergometer cycling have 66° of total knee motion [9]. Represented as the inner angle between the femur and the tibia, knee flexion reaches 134° at the bottom dead 180° crank position and 68° at the top 0° in the crank cycle. In general, however, different cyclists prefer different cycling positions [2]. Affected by lower seat height and more upright position, knee flexion values are usually lower when non-elite and especially non-road and commuter cyclists are considered, whereas higher values are expected for professionals, accustomed to a more aerodynamic position with drop bars. Although static bicycle fitting, considering joint angles at 180° and 0°, is a cheap and effective method, some researchers have indicated that further tuning of a cyclist's position could be done using dynamic measurements [4,10].

Optical motion tracking systems [11–13], goniometers [14], and video analysis [9,10,13–15] are most commonly used to dynamically measure joint angles during cycling. While all

these solutions provide for reliable measurements, their use is generally conditioned by a closed laboratory environment and cycling on the spot, using a stationary bicycle or a dedicated turbo trainer, on which the cyclist's personal bicycle is locked in place. Using mobile cameras is possible, but unfortunately it is mostly too expensive for regular or recreational use and for non-professional purposes.

More recently, motion tracking with video cameras is being replaced by wearable devices with integrated inertial measurement units (IMU), combining 3D accelerometers, gyroscopes, and magnetometers. Their small dimensions and lightness, coupled with low energy consumption, portability, availability, and ease of use are some of the reasons for their ubiquity and indispensability for tracking movements and movement patterns [16–23] and consequently for motion analysis in sports [24–28]. With the usage of different sensor fusion techniques, these devices have shown to be successful in orientation estimation and angle determination of different joints [29–33].

Since raw measurements of IMU sensors are prone to various errors, offset readings in particular, causing integration errors and in turn attitude and position estimates drift, IMUs are in general considered reliable for short measurement trials, measured usually in seconds [29–31] or up to a couple of minutes [32,33]. The accuracy for longer measurement trials is largely affected by the dynamic conditions of the motion captured.

Various error-compensation techniques are usually applied to improve results accuracy, the Kalman filter being one of the most popular and robust ones [30–33]. In the motion capture context, the Kalman filter relies on fusing signals from different inertial sensors and is especially used in dynamic environments. In [32] a magnetometer-free Kalman filter was designed for robust estimation of joint angles in magnetically disturbed environments. After conducting a set of two measurements, a short 1 min and another 3 min one, the values of the root mean square error (*RMSE*) were 1.88 and 3.04°, respectively. Additionally, magnetometers can be a valuable tool when combined with inertial sensors for absolute orientation estimation and motion tracking, and a more valuable insight on the matter is presented in [12,13,34–37]. Sensor fusion algorithms designed with magnetometer usage in mind, such as the Tilt Kalman filter and Madgwick filter, improve orientation estimation in many applications [38,39].

Given the benefits of IMUs, if such sensors prove reliable for measuring joint angles during cycling, they will benefit the industry. Cheaper and more accessible equipment will be developed and used in real-world environments. Consequently, such equipment will be useful to professional as well as recreational cyclists in improving their technique.

Some research has already been conducted in this field [12,13,40]. The author of [12] used a full MVN body suit [41], housing 17 IMU devices and relying on anthropometry and a biomechanical model to define and assemble individual segments and track kinematics during road cycling. The hip, knee, and ankle joint angles were obtained using the Kinematic Coupling algorithm, based on an error-state Kalman filter utilizing accelerometer, gyroscope, and magnetometer readings. For each of the 10 subjects participating, 3 short indoor measurements were evaluated using an optical system, each measurement being 1 min in length, with the cadence goal being between 90–110 rpm. *RMSEs* were in the ranges 0.8–0.9°, 3.1–3.4°, and 2.2–2.8° for the hip, knee, and ankle joint, respectively. Using the Madgwick filter, the authors of [13] report similar *RMSE* values for the knee angle, i.e., below 3.8° for short measurements, no longer than 30 s.

In [40] researchers measured the ankle joint angle with an IMU attached to the cyclist's pedal, using 3D accelerometer and gyroscope readings only, passed through a Kalman filter. Evaluation was done using two optical encoders, one on the ergometer pulley and another on the pedal. The *RMSE* of 10 short 45 s measurements on a single individual amounted to $2.77° \pm 0.10°$.

However, to the best of our knowledge, a more comprehensive research, investigating the accuracy of joint angle estimation using IMUs, considering short-term measurements as well as those longer in duration, is missing and would highly benefit the community.

We consider a simple method for measuring joint angles, relying on 3D accelerometer and gyroscope measurements from two IMU devices, tracking the rotations of two adjacent body parts, in turn measuring the angle of the connecting joint. Road bicycles, even carbon framed ones, contain parts made of ferrous materials such as the chain, chainring, sprocket, small metal gearboxes and brakes components, bolts and screws, felts with metal spikes, drive and steering bearings, and metal cleats of the clipless bicycle pedals, all rotating at a high speed characteristic to the cycling environment. The magnetic fields of these ferrous parts overpower the Earth's magnetic field causing the magnetometer readings to become unreliable [12]. Consequently, the usage of a magnetometer, and the previously mentioned algorithms, utilising this sensor, are in this study omitted. To evaluate the method, the knee joint is chosen. Measurements obtained using an optical motion tracking system are used for result evaluation. Two error compensation techniques are considered–simple gyroscope offset subtraction along with first-degree polynomial detrending and the Kalman filter.

We consider two durations of trials, 5 and 20 min. Providing for shorter-term accuracy evaluation, the former supports general position evaluation and bicycle fitting in dynamic conditions, mostly in closed, laboratory environments. Providing for longer-term accuracy evaluation, the latter could be used for further improvement of biological feedback loops, aimed towards improving a cyclist's technique, both during indoor and outdoor cycling. Additionally, longer-term measurement could be combined with other sensing technologies, e.g., for measuring muscle activity and power output, providing for deeper insight into the complexity of individual's biomechanics during cycling.

The presented research is a continuation of our preliminary investigation presented as a conference paper [42]. The methodology has been further developed and a more comprehensive evaluation has been conducted, including additional and longer measurement trials, obtained for an extended sample of subjects.

In all the subsequent sections, we use the following notation rules: large bold letters denote matrices, small bold letters denote vectors, and large or small italics denote scalars.

## 2. Materials and Methods

### 2.1. Determining Joint Angle from IMU Measurements

To measure the knee joint angle, one IMU device is placed on the subject's thigh and another on his or her shank, as illustrated in Figure 1. To avoid additional signal post-processing and transformations, one intrinsic coordinate system axis (in the presented Figure 1, the *x*-axis) of the IMU is aligned with the longitudinal axis of the respective segment, thigh or shank.

Denoting with $\alpha^{(IMU)}[n]$ the sought-after joint angle for each measurement step *n*, we can write:

$$\alpha^{(IMU)}[n] = \pi - \beta^{(IMU)}[n] - \gamma^{(IMU)}[n], \tag{1}$$

where $\beta^{(IMU)}[n]$ and $\gamma^{(IMU)}[n]$ are the angles of rotation for the thigh and the shank, respectively, determined using gravity reaction force vector projections on the *x* and *y* axes of each IMU. We denote the $1 \times 3$ vectors of these projections respectively for the thigh and shank IMU with $\mathbf{g}_1 = [g_{1_x}\ g_{1_y}\ g_{1_z}]$ and $\mathbf{g}_2 = [g_{2_x}\ g_{2_y}\ g_{2_z}]$. Considering the orientation of both sensors and simple geometry presented in Figure 1, we can write:

$$\beta^{(IMU)}[n] = \left| \arctan\left( g_{1_y}[n] / g_{1_x}[n] \right) \right|, \tag{2}$$

$$\gamma^{(IMU)}[n] = \arctan\left( g_{2_y}[n] / g_{2_x}[n] \right). \tag{3}$$

To determine the components of vectors $\mathbf{g}_1[n]$ and $\mathbf{g}_2[n]$ in (2) and (3) for each measurement step *n*, we first consider an initial stationary window of $N_s$ samples, preceding the actual cycling motion. By averaging the 3D accelerometer measurements during this window, we obtain $1 \times 3$ initial vectors $\mathbf{g}_1[0]$ and $\mathbf{g}_2[0]$. For each subsequent measurement sample, we rotate the current vector as measured by the 3D gyroscope. Denoting with $\mathbf{g}_i[n+1]$ and $\mathbf{g}_i[n]$ the subsequent and current iteration of the vectors $i = 1,2$, and with

$\mathbf{R}_i^{(gyro)}[n]$ the respective rotation matrix obtained from the gyroscope measurements, we can write:

$$\mathbf{g}_i[n+1] = \mathbf{g}_i[n]\mathbf{R}_i^{(gyro)}[n] .\tag{4}$$

The post-multiplication rotation order applied in (4) accounts for the fact that the 3D gyroscope measures rotations in its intrinsic, i.e., rotating coordinate system.

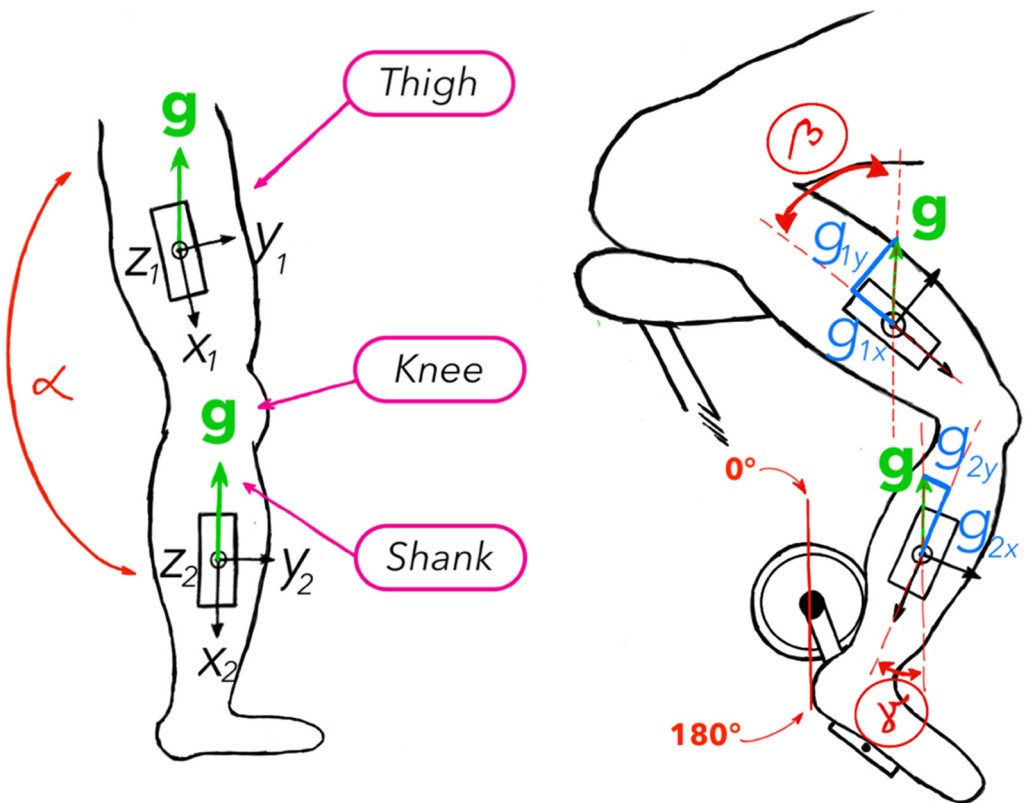

**Figure 1.** Placement of the IMUs on each body segment during standing position and during cycling.

In particular, the rotation matrix in (4) is obtained by considering the Simultaneous Orthogonal Rotation Angle (SORA), as presented in [43]. SORA correctly interprets the three rotations measured for each time sample *n* as occurring simultaneously. As long as the axis of rotation does not change during the sampling interval, SORA accurately represents the actual rotation of the gyroscope, around a single axis and angle. Accordingly, for each *n* the rotation axis is equal to the normalized angular velocity output of the gyroscope while the rotation angle is equal to the product of the angular velocity magnitude and the sampling interval. Denoting with $\boldsymbol{\omega}_i[n]$ a $1 \times 3$ vector of angular velocity measurements, $\mathbf{u}_i[n]$ and $\varphi_i[n]$ the axis and angle of rotation, respectively, and with $T_s$ the sampling interval, for each IMU $i = 1,2$ and measurement sample *n*, we can write:

$$\mathbf{u}_i[n] = \boldsymbol{\omega}_i[n]/|\boldsymbol{\omega}_i[n]|,\tag{5}$$

$$\varphi_i[n] = |\boldsymbol{\omega}_i[n]|\, T_s,\tag{6}$$

$$\mathbf{R}_i^{(gyro)}[n] = \begin{bmatrix} u_{ix}^2 + c\varphi_i\left(u_{iy}^2 + u_{iz}^2\right) & (1-c\varphi_i)u_{ix}u_{iy} - s\varphi_i u_{iz} & (1-c\varphi_i)u_{ix}u_{iz} + s\varphi_i u_{iy} \\ (1-c\varphi_i)u_{ix}u_{iy} + s\varphi_i u_{iz} & u_{iy}^2 + c\varphi_i\left(u_{ix}^2 + u_{iz}^2\right) & (1-c\varphi_i)u_{iy}u_{iz} - s\varphi_i u_{1x} \\ (1-c\varphi_i)u_{ix}u_{iz} - s\varphi_i u_{iy} & (1-c\varphi_i)u_{iy}u_{iz} + s\varphi_i u_{ix} & u_{iz}^2 + c\varphi_i\left(u_{ix}^2 + u_{iy}^2\right) \end{bmatrix}\tag{7}$$

In (7) $s\varphi$ and $c\varphi$ are additionally introduced and used as abbreviated notations respectively for the sine and cosine functions of the rotation angle $\varphi$.

Opting for using the rotation matrix (7) instead of the rotation quaternion in (4) is conditioned by our research aim of obtaining and evaluating orientation results for each time sample $n$. While the rotation quaternion would provide for equally valid results, the rotation matrix has a slight computational advantage when calculating orientation on a sample-by-sample basis [44]. If we were to calculate orientation not after each rotation but after a couple or more, the rotation quaternion would be used instead.

### 2.2. IMU Measurements Error Compensation and Sensor Fusion

Accelerometer and gyroscope sensors suffer from various sources of measurement errors, most considerably output bias [45,46]. Given this drawback, the orientation results calculated from the gyroscope measurements on an integration basis, are known to drift, especially for longer measurements. To compensate for this effect, we apply a simple calibration technique. We estimate the gyroscope output bias from a calibration stationary window of $N_c$ samples. We then subtract this offset from all subsequent measurement samples. While the $N_s$ window that is used for determining the initial values of vectors $\mathbf{g}_1[0]$ and $\mathbf{g}_2[0]$ is a relatively short window, occurring immediately prior to the cycling motion, the $N_c$ window is relatively longer and occurs prior to the measurement trial, i.e., prior to $N_s$ and the cycling motion. Finally, the $\mathbf{g}_1$ and $\mathbf{g}_2$ vectors are detrended with a polynomial of the first degree before the arctan function evaluation and $\beta^{(IMU)}[n]$, $\gamma^{(IMU)}[n]$, and $\alpha^{(IMU)}[n]$ estimation in (1)–(3).

Additionally, we apply the Kalman filter sensor fusion technique, combining accelerometer and gyroscope measurements to provide for better orientation estimates. Essentially, the filter exploits the fact that both the accelerometer and gyroscope measure orientation–the former operating as an inclinometer and measuring vector $\mathbf{g}$ projections on the sensor intrinsic coordinate system axes and the latter measuring angular velocities around those same axes.

In particular, we use an error-state Kalman filter [12,47–49], as implemented in [50]. The filter is fed with raw acceleration measurements and offset compensated gyroscope readings. The implemented filter then iteratively attempts to track errors in orientation, gyroscope output bias, and linear acceleration to output final orientation and angular velocity. For each iteration step, the filter first predicts orientation from the gyroscope readings, compensated for output bias and estimates the error in vector $\mathbf{g}$ projections, i.e., the difference between two calculated vectors $\mathbf{g}$–one by considering the predicted orientation and the other by considering the acceleration measurements, having the linear acceleration estimated in the previous iteration step subtracted. Using this error, the Kalman filter parameters are updated according to the Kalman equations and used to finally correct the predicted orientation, gyroscope output bias and linear acceleration.

For a properly functioning and stable Kalman filter, its properties, defining the observation model noise, must be appropriately tuned. The gyroscope noise parameter is set equal to the maximum variance of the gyroscope readings while stationary, whereas the gyroscope drift noise is set up to a low value, determined empirically. Similarly, accelerometer noise is set equal to the maximum variance of the accelerometer readings while stationary, while the linear acceleration noise, characterising the acceleration experienced by the sensor during the measurement, is set equal to the maximum variance of the accelerometer during the cycling part of the measurement. The linear acceleration decay factor parameter is set experimentally to best reflect the dynamic nature of the captured motion.

### 2.3. Determining Joint Angle from Optical Motion Tracking System Measurements

To evaluate the joint angle determined from IMU measurements, one rigid body per IMU is designed and tracked by the optical motion tracking system. Each rigid body is constructed using a flat brick $12.8 \times 4.7$ cm in size from The Lego Group, Billund, Denmark, holding three optical markers, together with the IMU device. The IMUs and the markers are fixated using two-side adhesive tape. The central marker of the rigid body is placed on the IMU itself. Using additional, smaller Lego® bricks the remaining two optical marker

are positioned to lie along the sensor's intrinsic $x$ and $y$ axis. All three markers define a coordinate system of the rigid body, aligned with that of the sensor.

The optical motion tracking system has reasonable difficulties recognizing similar rigid bodies at once, which results in occasional permutations of such bodies during the recognition process. To make the difference in appearance of the two rigid bodies more apparent and their recognition more accurate, one marker of each rigid body is additionally elevated for a predetermined height. This change in marker position causes a slight misalignment of the IMU and rigid body coordinate frames and is compensated by defining an additional, virtual marker in the optical motion tracking system software, at the position where the marker was originally placed and using it instead of the elevated marker for defining the rigid body's intrinsic coordinate system. The final distance between the central and the two outer markers is 9 and 3.5 cm for the thigh and 10.7 and 3.5 cm for the shank rigid body.

Both rigid bodies are fixated to the corresponding body segment using medical cohesive elastic bandage.

To get the crank position and determine the phase of the cycle, an additional marker is placed on the bottom of the bicycle pedal, just below its intrinsic axis of rotation.

With all the sensors attached, the cyclist cycles in a supervised environment on a road bicycle fixated on a turbo trainer. The overall measurement setup is illustrated in Figure 2.

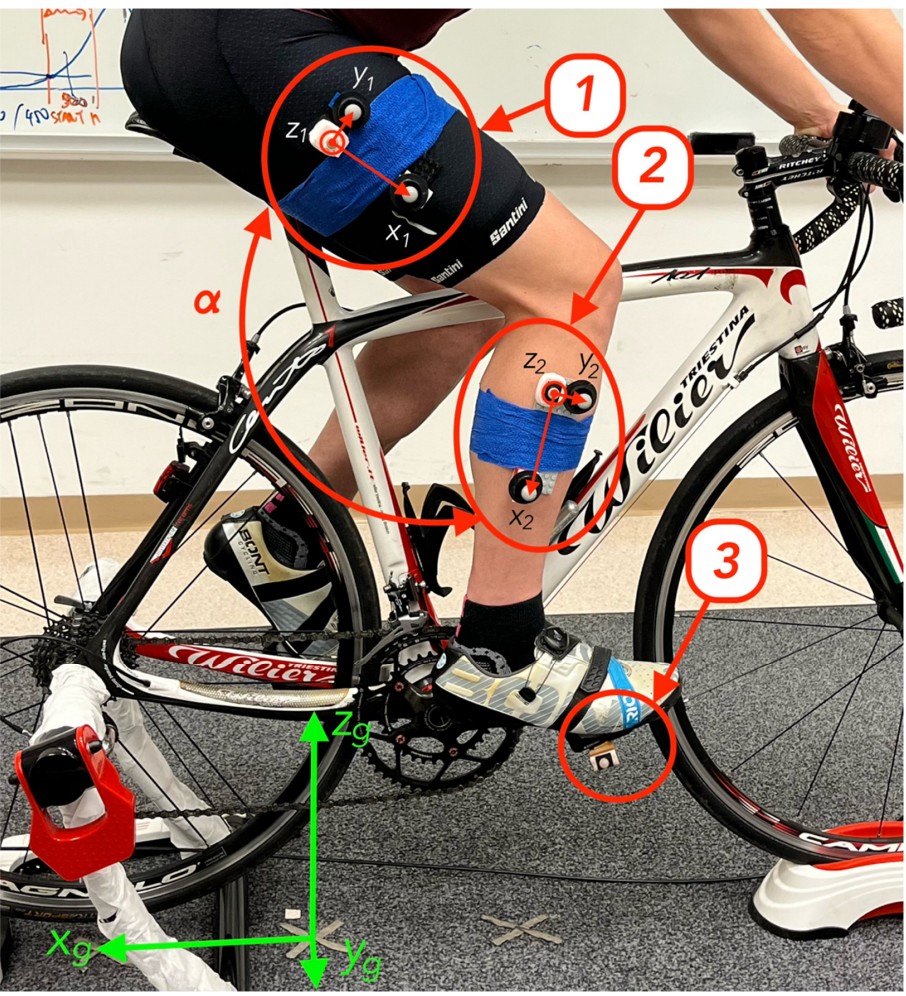

**Figure 2.** Experimental setup for measuring the knee joint angle. 1 and 2 denote the rigid bodies used for determining the knee joint angle with the optical motion tracking system, whereas 3 denotes the additional marker used for the crank angle measurements. The two IMU devices are positioned below the central points of the two rigid bodies.

For each measurement sample $n$, the optical tracking system gives the orientations of both rigid bodies $\mathbf{R}_1^{(ref)}[n]$ and $\mathbf{R}_2^{(ref)}[n]$ in the global frame. To determine the knee joint angle, a connecting rotation matrix, aligning the first rigid body with the second one and denoted with $\mathbf{R}_{conn}^{(ref)}[n]$ is calculated. We can write:

$$\mathbf{R}_{conn}^{(ref)}[n] = \left(\mathbf{R}_1^{(ref)}[n]\right)^{-1}\mathbf{R}_2^{(ref)}[n]. \tag{8}$$

The angle between the two orientation matrices $\mathbf{R}_1^{(ref)}[n]$ and $\mathbf{R}_2^{(ref)}[n]$, i.e., the knee joint angle, is then calculated as the angle of rotation described by $\mathbf{R}_{conn}^{(ref)}[n]$:

$$\alpha^{(ref)}[n] = acos\left[\left(tr\left(\mathbf{R}_{conn}^{(ref)}[n]\right) - 1\right)/2\right], \tag{9}$$

where tr denotes the trace function, i.e., the sum of all elements on the matrix diagonal.

If, due to numerical errors, the absolute value of the *acos* function argument in (9) exceeds 1, leading to a complex solution for $\alpha^{(ref)}[n]$, the angle of rotation is calculated using the *asin* function instead. Since we are interested in only the angle and not the direction of rotation, we use the absolute values of the asin function and calculate $\alpha^{(ref)}[n]$ according to:

$$\alpha^{(ref)}[n] = k\pi + (-1)^k\left|asin\left[\sqrt{\left(\mathbf{R}_{conn}^{(ref)}[n]_{3,2} - \mathbf{R}_{conn}^{(ref)}[n]_{2,3}\right)^2 + \left(\mathbf{R}_{conn}^{(ref)}[n]_{1,3} - \mathbf{R}_{conn}^{(ref)}[n]_{3,1}\right)^2 + \left(\mathbf{R}_{conn}^{(ref)}[n]_{2,1} - \mathbf{R}_{conn}^{(ref)}[n]_{1,2}\right)^2}/2\right]\right|, \tag{10}$$

where $k$ is equal to 1 if the *acos* argument in (9) is negative and 0 otherwise, extending the *asin* function evaluation in (10) to also support results between $\pi/2$ and $\pi$.

### 2.4. Alignment of IMU and Optical Motion Tracking System Coordinate Frames

Despite best efforts one can invest in the physical alignment of the rigid body coordinate system with that of the IMU, the two will, in general, never exactly match. The human-introduced error during marker positioning, the use of supportive base and adhesive tape to fixate the markers all contribute to a systematic orientation misalignment.

Even in the highly improbable case of perfectly positioned optical markers, used to define the rigid body coordinate system, the markers themselves measure 1.1 centimetre in diameter, giving way to coordinate system definition uncertainty. A misalignment of the rigid body's coordinate system by a few degrees is hence expected and should be accounted for to provide for relevant result evaluation.

We experimentally determine a rotation matrix, defining the orientation of the rigid body coordinate system relative to that of the respective IMU's. We perform a short measurement, starting from a stationary state ($s = 0$) in an arbitrary orientation and eventually rotating both rigid bodies into three additional distinct arbitrary orientations ($s = 1,2,3$). For each rigid body $i$ we then search for a rotation matrix that best aligns the IMU and optical system measurements.

We compensate the gyroscope measurements for output bias. Further using the rotation axis and angle according to (5) and (6), we calculate the four orientations of the IMU in its initial coordinate frame. We denote the obtained orientations for each rigid body $i$ with $\mathbf{O}_i^{(gyro)}[s]$, where $s$ denotes all four arbitrary states, i.e., $s = 0,1,2,3$. Note that, since we are representing all four orientations with respect to the initial state ($s = 0$), the following holds $\mathbf{O}_i^{(gyro)}[0] = \mathbf{I}$, where $\mathbf{I}$ denotes a $3 \times 3$ identity matrix.

We then consider the optical system orientation measurements, denoted with $\mathbf{O}_i^{(ref)}[s]$. Since $\mathbf{O}_i^{(ref)}[s]$ orientations are given in the global frame of the optical motion tracking system, we first perform the following transformation:

$$\left(\mathbf{O}_i^{(ref)}[0]\right)^{-1}\mathbf{O}_i^{(ref)}[s], \tag{11}$$

to obtain the four orientations representations in the initial system of the rigid body *i*. Since both the IMU and the optical motion tracking system should measure the same orientations in their respective intrinsic coordinate systems, without measurement errors and misalignment, $\mathbf{O}_i^{(gyro)}[s]$ would match (11). Since the measurements are short and gyroscope bias is compensated for, we attribute the difference in the measured orientation entirely to the IMU and rigid body misalignment. For each *i* we then search for a rotation matrix $\mathbf{R}_{miss\,i}$ that best aligns $\mathbf{O}_i^{(gyro)}[s]$ and $\mathbf{O}_i^{(ref)}[s]$, i.e., that best fits the following expression:

$$\mathbf{R}_{miss\,i}(\mathbf{O}_i^{(ref)}[0]\,)^{-1}\mathbf{O}_i^{(ref)}[s]\mathbf{R}_{miss\,i}^{-1}\ = \mathbf{O}_i^{(gyro)}[s] \tag{12}$$

for *s* = 1,2,3. We achieve this using a brute force method–by composing all rotation matrices possible $\mathbf{R}_{miss\,i}$, for a rotation angle between 0 to 5°, with 0.05° resolution, and for all possible rotation axes with a 0.05 resolution for *x*, *y*, and *z* projections in the range [−1,1]. Considering each such composed matrix, we calculate the difference matrix:

$$\Delta_i[s] = \boldsymbol{O}_i^{(gyro)}[s] - \mathbf{R}_{miss\,i}(\boldsymbol{O}_i^{(ref)}[0]\,)^{-1}\boldsymbol{O}_i^{(ref)}[s]\ \mathbf{R}_{miss\,i}^{-1} \tag{13}$$

for *s* = 1,2,3. Finally, $\mathbf{R}_{miss\,i}$ for which the following criteria:

$$\mathrm{argmin}\left(\sum_{s=1}^{3} sumsq(\Delta_i[s])\right) \tag{14}$$

is met, where *sumsq* denotes the sum of matrix elements' squares, is set as the matrix of misalignment between the rigid body *i* coordinate system and the respective IMU.

Using $\mathbf{R}_{miss\,i}$ all optical system measurements $\mathbf{R}_i^{(ref)}[n]$ are compensated according to:

$$\mathbf{R}_{miss\,i}\mathbf{R}_i^{(ref)}[n]\mathbf{R}_{miss\,i}^{-1} \tag{15}$$

and used instead of the original $\mathbf{R}_i^{(ref)}[n]$ (8) values for calculating the reference knee joint angle in (9)–(10).

## 2.5. Time-Synchronizing Joint Angle Measurements

The IMU and optical system joint angle results are synchronised considering the correlation function between the first 5 s of their run (including the stationary period and at least the first two periods) and using the position of this function's maximum as the synchronisation time-shift value.

Since no two systems have perfectly matching clocks, for longer measurements, despite the afore-mentioned mutual time-synchronisation, the IMU and optical system angles desynchronise with ever more divergence as the angles run longer. This effect is illustrated in Figure 3. Despite the initial time-synchronisation, after 124,000 samples (slightly over 20 min for a 100 Hz sampling frequency), the results are misaligned in time for 7 samples on average.

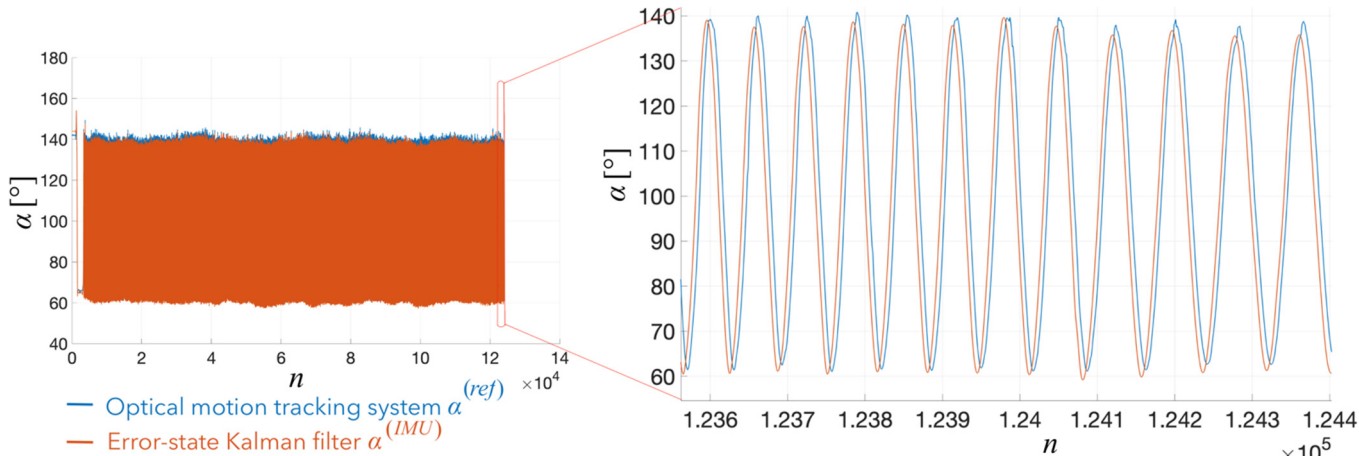

**Figure 3.** Desynchronisation of results for longer measurement trials due to systems' clock-differences.

To compensate for this effect the Longest Common Subsequence (LCSS) algorithm is applied [51], removing samples from $\alpha^{(IMU)}[n]$ and $\alpha^{(ref)}[n]$ to obtain best time matching between both angles, as illustrated in Figure 4. The algorithm reduces the sampling frequency by a small margin, however, it efficiently removes the described systematic synchronization error. The maximum allowed jump between two samples was set to 6, corresponding to slightly less than 10% of the number of samples obtained during one crank cycle at 90 rpm cadence and 100 Hz sampling, i.e., 66.67 samples/cycle.

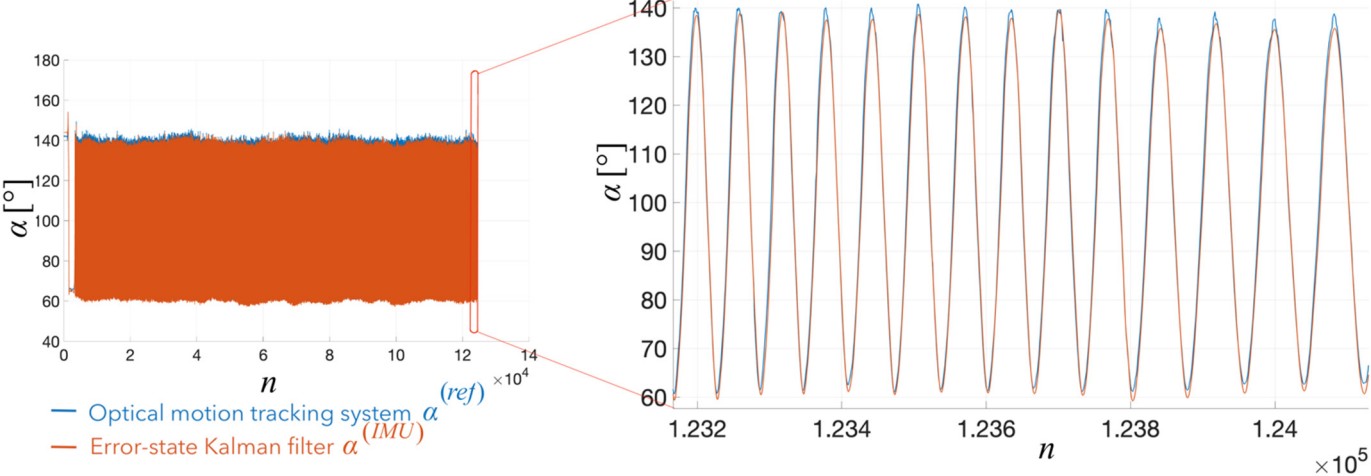

**Figure 4.** Synchronisation of results for longer measurement trials after using the LCSS algorithm.

### 2.6. Determining the Crank Angle

The crank angle, denoted with $\psi[n]$, is obtained from the coordinates of the pedal marker, as illustrated in Figure 2 denoted with 3, according to:

$$\psi[n] = atan2\left(\left(\left(x_3[n] - x_{3\,offset}\right)\right) / \left(z_3[n] - z_{3\,offset}\right)\right),\tag{16}$$

where $x_3[n]$ and $z_3[n]$ are the marker $x$ and $z$ coordinates for each $n$ and $x_{3\,offset}$ and $z_{3\,offset}$ are the crank axis offsets from the global coordinate system origin in the $x$ and $z$ dimension, respectively.

## 3. Experimental Validation

### 3.1. Hardware

For the experiment two MetaMotionR inertial measurement units were used from Mbientlab Inc. [52], chosen for their small size, sampling frequency range, and ready-made software for simple logging and exporting of the conducted measurements. For the optical motion tracking system, the Qualisys Oqus 3-series from Qualisys AB, Gothenburg, Sweden, [53] was used, which consists of a set of infrared cameras positioned appropriately on the ceiling of the laboratory to capture the entire workspace.

The sampling frequency $f_s$ of both sensor systems was set up to 100 Hz, which was a safe compromise with respect to the frequency content of the captured motion and any future real-time transmission and processing needs.

### 3.2. Subjects

During a three-week period, 7 subjects participated in the study: 3 were females of the age between 32 and 39, whereas the 4 remaining subjects were males of age between 20 and 48. Each of the subjects gave a short description of their cycling experience along with their consent to conduct the said measurements. The subjects were asked to use their personal road bicycles if available, with seat height adjusted according to personal preference. If subjects did not have access to a personal bicycle, a substitute of proper size was obtained, adjusted seat-wise and used instead.

### 3.3. Measurement Protocol

With the workspace calibrated for the optical motion tracking system, the bicycle along with the turbo-trainer were placed so that the wheelbase of the bicycle aligned with the global $x$ axis, the bicycle stack with the global $z$ axis, and the handlebar width with the remaining $y$ axis of the global coordinate frame as denoted by $x_g$, $y_g$, and $z_g$ in Figure 2. After the rigid bodies with the IMUs were fixated on the subject, they were given enough time on the bicycle to get used to the setup and adjustments were made if any were necessary.

The subject was first asked to cycle for 5 min, with a stationary minute before. After sufficient rest, the subject was asked to cycle for another 20 min, with a stationary minute prior. A cadence consistency goal was set to 90 rpm, forced using a software metronome.

### 3.4. Signal Processing

Once the measurements were completed, they were all exported and further processed offline in MathWorks MATLAB R2021B [54].

Considering the associated timestamp values, signals from both IMU devices were mutually synchronised and resampled, to provide for common measurement samples at exact $T_s = 1/f_s = 0.01$ s time distance.

From the stationary minute, preceding both measurement trials, the last 20 consecutive samples determined to be stable for both IMUs were used for the $N_\mathbf{s}$ window to estimate the initial gravity reaction force vector projections, i.e., $\mathbf{g}_1[0]$ and $\mathbf{g}_2[0]$. The first following sample was set as the start of the measurement, i.e., $n = 1$. Additionally, from this stationary minute, for each IMU, 650 consecutive samples with the least angular velocity variance were used for the $N_c$ window for gyroscope offset compensation.

The IMU and optical motion system signals were then processed as presented in Section 2. In particular, for the Kalman filter parameters, the following were set: gyroscope drift noise = $2 \cdot 10^{-9}$ and linear acceleration decay factor = 0.25.

Anonymised data and all developed processing software are available for download from a publicly available repository [55].

## 4. Results

To evaluate the measured knee joint angles, the results are sliced up into cycles, using the angle's peak values, as illustrated in Figure 5. The interval from one peak value to

the next corresponds to one crank cycle. Table 1 summarizes the average cadence along with the maximum and minimum knee joint angles, as measured by the optical system, for each subject. For the 20 min measurement trial for Subject 5, the results are missing due to sporadic occlusions of the optical markers, causing the optical system to not only loose track but also confuse the markers when performing recognition, leading to incorrect application of the associated rigid body's coordinate frame.

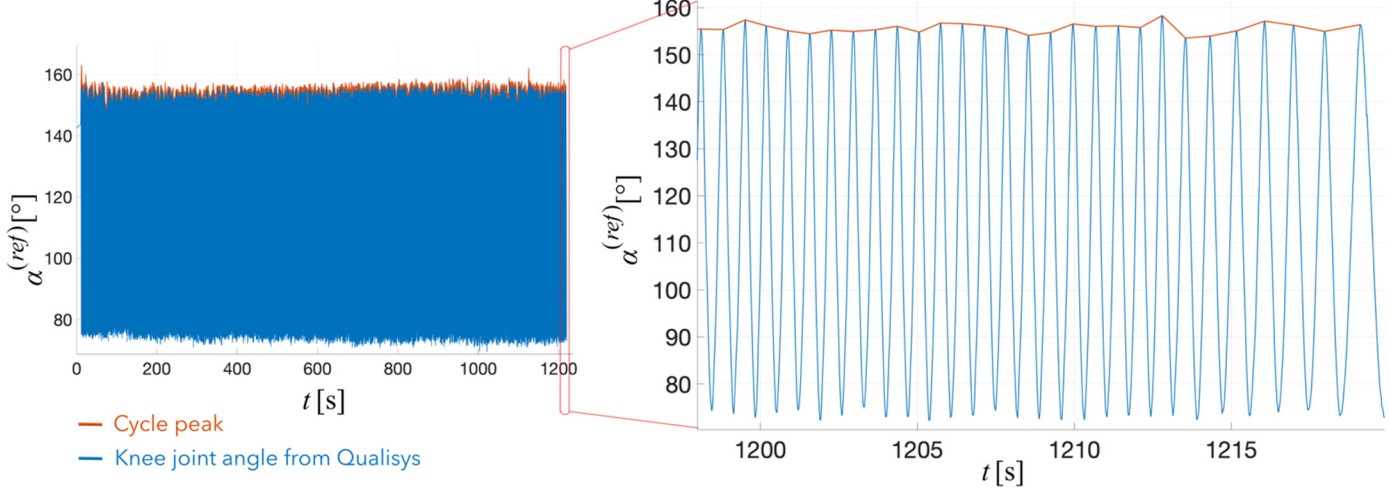

**Figure 5.** Example knee joint angle results. Maximum values are used to split the measurements into cycles. The cycles are then used for cadence, maximum angle, and average *RMSE* computation.

**Table 1.** The subjects' average cadence with the maximum and minimum knee joint angles as measured with the reference optical motion tracking system.

| Subject | Type of Measurement Trial | Average Cadence [rpm] | Max Knee Joint Angle [°] | Min Knee Joint Angle [°] |
|---|---|---|---|---|
| 1 | 5 min | $79.25 \pm 8.41$ | $141.31 \pm 7.57$ | $73.87 \pm 3.61$ |
|   | 20 min | $81.94 \pm 5.42$ | $146.97 \pm 4.79$ * | $74.56 \pm 2.44$ |
| 2 | 5 min | $90.50 \pm 5.70$ | $141.19 \pm 2.79$ | $62.55 \pm 0.96$ |
|   | 20 min | $89.68 \pm 6.28$ | $141.21 \pm 4.66$ | $59.52 \pm 4.29$ |
| 3 | 5 min | $68.45 \pm 6.76$ | $160.38 \pm 4.08$ | $76.86 \pm 1.96$ |
|   | 20 min | $88.73 \pm 8.91$ | $158.08 \pm 2.96$ | $76.39 \pm 2.67$ |
| 4 | 5 min | $86.51 \pm 5.49$ | $149.84 \pm 4.12$ | $78.88 \pm 2.70$ |
|   | 20 min | $88.68 \pm 9.86$ | $148.55 \pm 3.39$ | $75.07 \pm 3.89$ |
| 5 | 5 min | $95.03 \pm 7.18$ | $164.03 \pm 1.68$ | $81.90 \pm 1.07$ |
|   | 20 min ** | - | - | - |
| 6 | 5 min | $92.64 \pm 6.26$ | $132.94 \pm 3.67$ | $65.48 \pm 1.56$ |
|   | 20 min | $89.83 \pm 5.16$ | $130.57 \pm 3.32$ | $63.95 \pm 1.62$ |
| 7 | 5 min | $96.31 \pm 7.44$ | $138.83 \pm 3.00$ | $60.78 \pm 1.49$ |
|   | 20 min | $98.88 \pm 7.32$ | $139.63 \pm 3.00$ | $69.19 \pm 1.53$ |

* Bicycle seat height was additionally adjusted for subject's comfort after the 5 min trial, resulting in a noticeable increase in the max knee joint ** No results are given due to optical marker occlusion, resulting in no reference.

The average cadence was computed by averaging the number of samples for each cycle, i.e., angle peak value, and the cadence consistency goal of 90 rpm was best achieved by subjects 2, 4 and 6, whereas the rest of the subjects were either slightly quicker or faster, depending on the measurement trial.

For each cycle, angle *RMSE* is calculated using the optical motion tracking system's measurements as reference values. From the computed *RMSE* by cycle, the average *RMSE*

is calculated and presented in Table 2 as the final evaluation measure for each measurement trial and subject. The average *RMSE* for the last 10 cycles is also presented per each subject and measurement trial for comparison purposes. Additionally, the average *RMSEs* through all the 5 and 20 min measurement trials are given at the end of Table 2 for all *RMSE* categories.

**Table 2.** Knee joint angle average root mean square errors (*RMSE*) by crank cycle with an interval of ± two standard deviations for the entire measurement trial and the last 10 cycles. The average *RMSE* for each method is presented at the end of the table.

| Subject | Type of Measurement Trial | RMSE | | | |
| --- | --- | --- | --- | --- | --- |
| | | First-Degree Polynomial Drift Compensation [°] | | Kalman Filter [°] | |
| | | Entire Trial | Last 10 Cycles | Entire Trial | Last 10 Cycles |
| 1 | 5 min | 1.78 ± 1.31 | 2.77 ± 0.59 | 2.73 ± 0.92 | 2.62 ± 0.47 |
| | 20 min | 7.80 ± 7.69 | 15.52 ± 1.96 | 2.41 ± 0.91 | 2.29 ± 0.73 |
| 2 | 5 min | 7.84 ± 7.19 | 14.41 ± 0.30 | 1.45 ± 1.12 | 1.98 ± 0.32 |
| | 20 min | 12.58 ± 7.34 | 15.30 ± 0.41 | 1.46 ± 1.01 | 1.72 ± 0.32 |
| 3 | 5 min | 3.31 ± 1.44 | 4.31 ± 0.74 | 2.78 ± 1.09 | 2.92 ± 0.28 |
| | 20 min | 7.15 ± 5.12 | 10.76 ± 4.86 | 2.76 ± 0.81 | 2.42 ± 0.32 |
| 4 | 5 min | 2.88 ± 0.97 | 2.86 ± 0.69 | 2.66 ± 0.77 | 2.54 ± 0.76 |
| | 20 min | 23.24 ± 18.70 | 30.00 ± 4.92 | 2.58 ± 0.82 | 2.59 ± 0.80 |
| 5 | 5 min | 3.51 ± 1.00 | 4.62 ± 0.45 | 3.12 ± 0.46 | 3.12 ± 0.40 |
| | 20 min * | - | - | - | - |
| 6 | 5 min | 3.40 ± 1.61 | 3.80 ± 0.60 | 1.60 ± 0.75 | 2.21 ± 0.48 |
| | 20 min | 10.05 ± 12.15 | 22.70 ± 1.07 | 2.63 ± 0.70 | 2.36 ± 0.44 |
| 7 | 5 min | 1.30 ± 0.79 | 0.93 ± 0.25 | 1.39 ± 0.96 | 1.33 ± 0.34 |
| | 20 min | 9.64 ± 9.40 | 18.29 ± 1.09 | 1.39 ± 1.18 | 1.60 ± 0.25 |
| Average | 5 min | 3.45 ± 4.98 | 4.81 ± 8.23 | 2.18 ± 1.63 | 2.39 ± 1.21 |
| | 20 min | 11.75 ± 15.36 | 18.76 ± 12.77 | 2.18 ± 1.47 | 2.16 ± 0.90 |

* No results are given due to marker occlusion resulting in no reference.

*RMSE* of the knee joint angle calculated per cycle for all 5 and 20 min measurement trials are presented in Figures 6 and 7, respectively. All subjects performed at least 342 for the 5 min and 1630 cycles for the 20 min measurement trials. Measurement trials for which the average *RMSE* values for the entire trial, as presented in Table 2, do not exceed the average *RMSE* of, respectively, all 5 or 20 min measurement trials for the Kalman filter are coloured blue while the remaining measurements trials are coloured red. The average entire trial *RMSE* obtained using the Kalman filter was chosen for comparison due to being the lowest between the two total averages.

From the results presented in Figures 6 and 7 we can observe a non-negligeable difference between the knee joint angle values, measured with the two systems, already at the beginning of the measurements. This might come as a slight surprise, since the IMUs' and rigid bodies' coordinate systems are additionally initially aligned. Detailed investigation has shown that for the first couple of samples, the knee joint angles difference between both systems is indeed negligeable; however, calculating the RMSE per cycle basis, is affected by integration errors already for the first cycle, leading to noticeable difference from the beginning of the measurements.

Finally, Figure 8 presents the knee joint angles, $\alpha^{(\text{IMU})}$ and $\alpha^{(\text{ref})}$, with respect to the crank angle $\psi$, where the latter is computed from the added marker below the centre of the bicycle pedal, for one example 5 min and one example 20 min measurement.

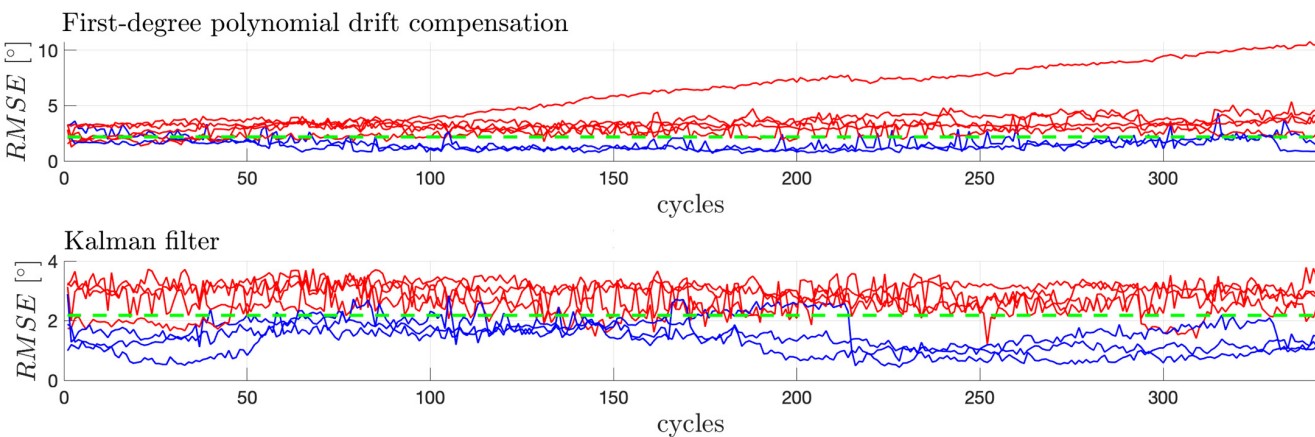

**Figure 6.** Knee joint angle root mean square error (*RMSE*) by crank cycle for all conducted 5 min measurements. The green dashed line represents the average entire trial *RMSE* of the Kalman filter calculated across all cycles and all subjects. The red lines represent measurements for which the average *RMSE* per cycle exceeds the average *RMSE* of the Kalman filter.

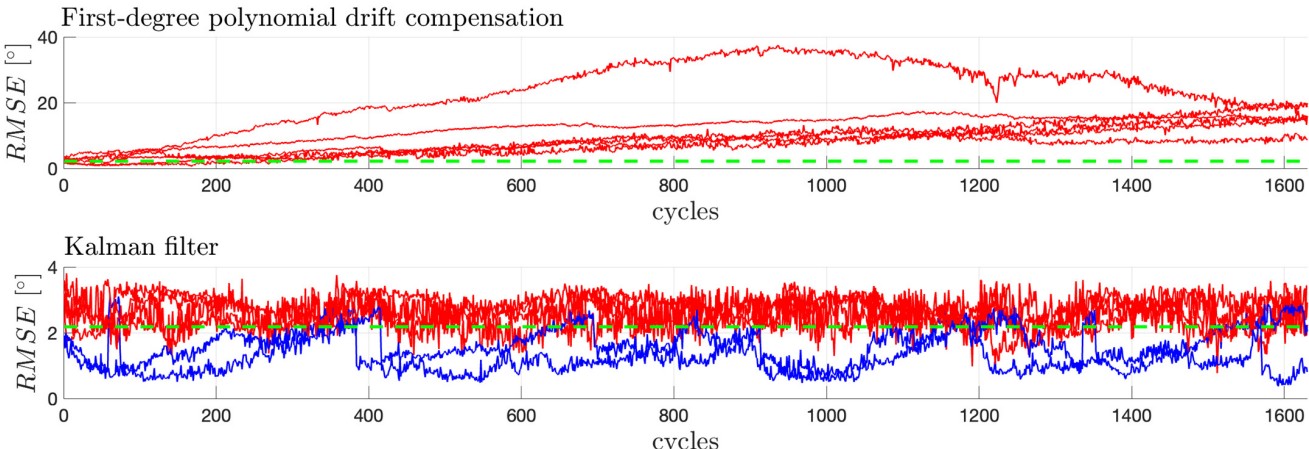

**Figure 7.** Knee joint angle root mean square error (*RMSE*) by crank cycle for all conducted 20 min measurements. The green dashed line represents the average entire trial *RMSE* of the Kalman filter calculated across all cycles and all subjects. The red lines represent measurements for which the average *RMSE* per cycle exceeds the average *RMSE* of the Kalman filter.

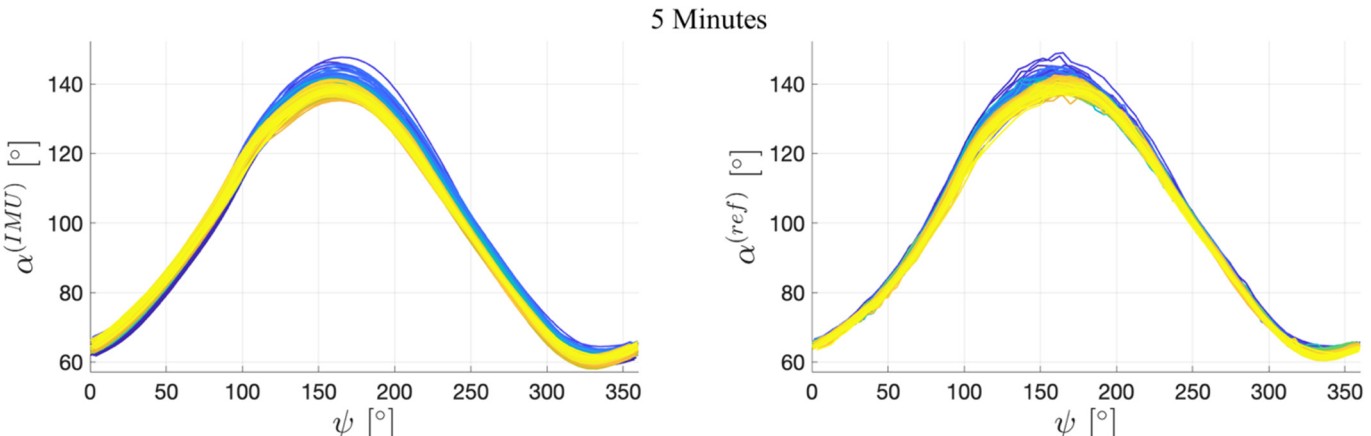

**Figure 8.** *Cont.*

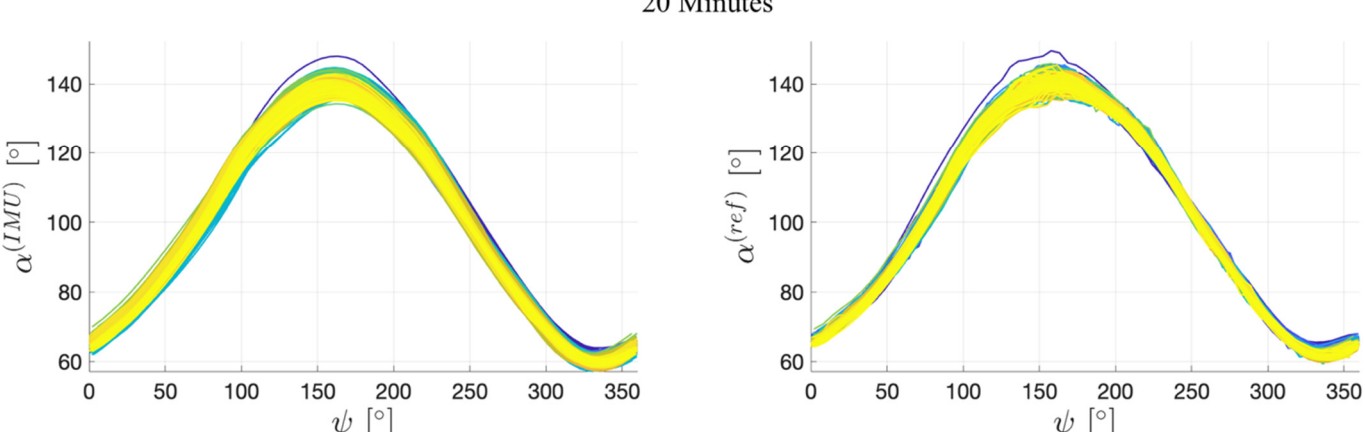

**Figure 8.** Knee joint angles obtained using the proposed IMU-based method and reference optical motion tracking system, respectively plotted with respect to the computed crank angle. The gradient from dark blue to bright yellow is used to differentiate subsequent cycles, from the first to the last.

## 5. Discussion

From the *RMSE* values presented in Table 2, we can observe that the Kalman filter clearly provides for better knee joint angle results, with the average *RMSE* values per cycle being in-between 1.39 and 3.12° for all 5 min measurement trials. For comparison, the average *RMSE* values per cycle for all 5 min measurement trials for the first-degree polynomial drift compensation method are in-between 1.30 and 7.84°.

The *RMSEs* values obtained for the 5 min trials for the Kalman filter are, in general, slightly lower than those presented in literature so far (3.1–3.4° for 5 min measurements in [12] and below 3.8° for 30 s measurements in [13]). We can attribute this improvement to offset subtraction and to additional initial alignment of the gyroscope and optical system rotation matrices.

Further on, we can observe that for the 20 min measurement trials, the knee joint angles *RMSE* significantly increases for the first-degree polynomial drift compensation method, with *RMSE* values being now in-between 7.15 and 23.24°. The accuracy for the Kalman filter method is practically unaffected, with *RMSE* values being now in-between 1.39 and 2.76°. To put into perspective, as the measurement duration increases for 300%, the average *RMSE* for the Kalman filter remains the same. This is even more evident when comparing the measurement average *RMSE* and the average *RMSE* for only the last 10 cycles, where the latter's average *RMSE* is comparable to the average trial *RMSE*. In contrast, the average *RMSE* for the last 10 pedalling cycles for the first-degree polynomial drift compensation method is significantly higher than the average trial *RMSE* for almost all of the measurement trials.

The results presented confirm the hypothesis that the measurement error due to IMU limitations increases with the measurement duration and that this error can be significantly decreased with appropriate error compensation techniques. Given the results presented in Figures 6 and 7, we can observe that while for some 5 min measurements, the *RMSE* values per cycle for the first-degree polynomial drift compensation method are comparable to those obtained for the Kalman filter method, the results for all 20 min measurements are significantly worse. In addition, the *RMSE* per cycle obtained for the first-degree polynomial drift compensation method is well above the total measurement average *RMSE* obtained for the Kalman filter after just a few minutes. In contrast, the Kalman filter results are below the given margin for the vast majority of time, confirming results stability as is expected from an adaptive solution.

To note, detrending of the measurements with a polynomial of the second degree was considered, assuming the gyroscope offset a linear process, leading to angle errors with a quadratic drift characteristic. However, this approach gave better results only for longer

measurements and even then, the improvement was only on a case-by-case basis during preliminary testing, its use was omitted. In contrast, all of the conducted measurements benefited from the first-degree polynomial detrending, assuming linear angle drifts. In addition, preliminary testing also showed that using the complementary filter, due to the highly dynamic characteristic of the captured cycling motion, does not improve results and was hence omitted.

Further on, initial measurements with the IMU positioned on the subject's shank have shown up to 30% deviation in magnetic field intensity when the subject is pedaling with a cadence of 90 rpm as oppose to when he is just sitting on the bicycle. In addition, this increase is a function of rotation speed so any variation in cadence would introduce further errors. As a result, further improvements gained through the usage of a magnetometer were deemed unlikely, and its use was omitted altogether.

Comparing the achieved accuracy of joint angles to the long-term accuracy of general orientation, obtained from robust single IMU orientation estimators, shows that the method presented either achieves or surpasses accuracies presented previously in literature [36,37]. Considering 1° *RMSE* on average achieved per Euler angle, roll and pitch (without considering the heading information), and per sensor as presented in [36], via adaptively tuning the measurement noise covariance matrix, amounts to approximately 2° for the full 3D joint angle, estimated using two IMUs, which is comparable to the average 2.18° *RMSE* obtained in this study. In [37] the reported median *RMSE* values per Euler angle obtained using an extended Kalman filter range from 3.8 to 7.8° for long-term measurements with various dynamic activities performed. The reason behind the fact that *RMSE* values obtained in this study are significantly lower than those presented in [37] stems from the specific cycling measurement environment in which the cycling cadence and motion are constrained. In such a case, the Kalman filter parameters do not have to be adapted during measurements and using constant values already gives highly accurate results. On the contrary, the method presented in [37] targets to achieve robust orientation estimation in various dynamic and static environments and activities. Using robust orientation estimation methods as presented in [36,37] is however expected to improve result accuracy for outdoor measurements, during which the cyclist, while still aiming to maintain a constant cadence, occasionally slows down or, for short-term recovery, stops pedalling altogether. Further investigations in this area would be highly beneficial for in-depth cycling biomechanics analysis.

While the presented results are highly promising, it is also important to note that significant attention was devoted to correctly aligning the rigid bodies with each respective body part, i.e., the thigh and the shank. A slight misalignment of the IMU from the body segment would introduce a discrepancy between the de facto measured angle and the target angle. For day-to-day measurements, when measurement preparation time should be optimized, this could be accounted for by adopting a segment to segment based IMU calibration technique, such as the one presented in [13].

Finally, comparing the Kalman IMU and reference knee joint angles with respect to the crank angle as can be seen in Figure 8, both follow the same trend with the IMUs measurements having only a slightly higher variance in comparison, observed as total thickness of the graph once all cycles are drawn, especially at crank position 0°. Additionally, in the same Figure we can observe that the optical motion tracking systems is prone to sporadic instantaneous errors, while the IMU measured angle has a smoothed run, as is expected in the given cycling dynamic conditions. This confirms the notion that the knee joint angle computed from IMUs can be reliably used with respect to the crank angle. Furthermore, this opens the door to its combination with data obtained using other sensor devices such as the electromyogram for tracking of muscle activation patterns.

## 6. Conclusions

Given the results obtained, the presented method could be used for precision fitting for professional cyclists. Instead of using expensive optical systems and other various

methods, the optimal angles, not only the knee joint, but others as well, such as the hip joint angle, computed with IMUs could be used as a replacement. This would in turn lead to more affordable equipment for professional and amateur cyclists alike.

Extending the investigation from the controlled laboratory environment, the next step would be to confirm whether the presented solution proves equally reliable in outside world environments, where the presence of uneven road surfaces and corners could introduce challenges, probably requiring additional signal filtering techniques and robust joint angle estimators. In addition, inclusion of magnetometer readings could prove useful for tracking bicycle heading in outside environments, provided it is not mounted near the rotating ferrous bicycle parts, compromising the Earth's magnetic field measurements.

Furthermore, the method presented in this paper could be used not only for measuring the knee joint angle, but for other active joints during cycling. Measuring the hip joint angle however in particular would also require additional signal filtering techniques, since one IMU would have to be placed in the sagittal plane of the right lumbar region where it would be affected by subject's breathing, making the method less stable in the process.

The ever-growing field of technique monitoring and improvement with biofeedback loops for cyclists could also benefit from the presented method. Even without the actual evaluation of joint angles reliability, some promising research has been presented in this field using IMUs. For example, the authors of [56] developed a method for aligning the cyclist motion trajectories with previously recorded professional's motion cycles and in [57] optimal cycling profile guidance has been presented, both relying on IMU joint angle measurements.

Finally, combining the results obtained from the presented method with readings from other sensor technologies, e.g., for muscle activation and contraction monitoring, could provide cyclists with crucial information on which muscle groups to focus on at a certain moment, potentially leading to more proficient muscle usage. This combined information could be given to the cyclist visually using a display on the bicycle handlebar, through sound with headphones, or haptically through vibrating actuators positioned appropriately on the cyclist's body.

**Author Contributions:** Conceptualization, S.O. and S.S.; methodology, S.O. and S.S.; software, S.O. and S.S.; validation, S.O. and S.S.; writing – original draft, S.O. and S.S.; writing – review and editing, S.O. and S.S.; visualization, S.O.; supervision, S.S. All authors have read and agreed to the published version of the manuscript.

**Funding:** This research was funded by the Slovenian Research Agency, grant number P2-0246 ICT4QoL—Information and communication technologies for quality of life.

**Institutional Review Board Statement:** This study was conducted for time periods of up to 2 h in length with the participants performing training activities expected of amateur cyclists. All of the participants gave their informed consent prior to conducting the measurements. The study was approved by the University of Ljubljana Human Research Ethics Committee (KERL UL) (No. 025-2022) and followed the Code of Ethics of the University of Ljubljana, which provides guidelines for studies involving human beings and is in accordance with the Declaration of Helsinki.

**Informed Consent Statement:** Informed written consent was obtained from all subjects involved in the study.

**Data Availability Statement:** The anonymised data is freely available from a public repository or by request through e-mail.

**Conflicts of Interest:** The authors declare no conflict of interest. The funders had no role in the design of the study; in the collection, analyses, or interpretation of data; in the writing of the manuscript, or in the decision to publish the results.

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
