# Peer review of "Inertial Measurement Units’ Reliability for Measuring Knee Joint Angle during Road Cycling"

_electronics, doi:10.3390/electronics12030751_

Round 1

Reviewer 1 Report

Why didn't we use the magnetic value in addition to the acceleration and angular velocity to estimate the angle?

Considering these factors, angle estimation can be expected to produce better results.

Sabatini, A. M. (2011). Kalman-filter-based orientation determination using inertial/magnetic sensors: Observability analysis and performance evaluation. Sensors, 11(10), 9182-9206.

Algorithms for comparing accuracy of angle estimation are not suitable.

I think it should be compared with popular algorithms such as the Tilt Kalman filter, and Madgwick algorithm.

Yean, S., Lee, B. S., Yeo, C. K., & Vun, C. H. (2016, October). Algorithm for 3D orientation estimation based on Kalman filter and gradient descent. In 2016 IEEE 7th Annual Information Technology, Electronics and Mobile Communication Conference (IEMCON) (pp. 1-6). IEEE.

I have a question about novelty.

Although it is the first time to estimate the joint angle, looking at the algorithm, the posture of IMU (1, 2, 3) is estimated individually by general sensor fusion, and the angle is calculated.

I think the error depends on the error of each individual IMU, is this interpretation correct?

In that case, again, comparisons with studies that robustly estimate angles with a single IMU (rather than the first-order polynomial correction equation) are needed.

Reviewer 2 Report

The paper shows moderate originality, based on the possibility to make inertial sensor measurements for a long time compensating drift.

In my experience, I have used inertial sensors with a digital motion processor (DMP) which compensates offset and drift, but this is another matter.

However, to my knowledge, use of the Rotation Matrix instead of Quaternions can produce "gimbal lock". Modern practice is to avoid the use of gimbals entirely, that can be done by integrating sensed rotation and acceleration digitally using quaternion methods to derive orientation and velocity.

doi of ref [38] is wrong

Round 2

Reviewer 1 Report

I have verified that the resubmitted manuscript has been properly revised.